

# Identification of *Meloidogyne panyuensis* (Nematoda: *Meloidogynidae*) infecting Orah (*Citrus reticulata* Blanco) and its impact on rhizosphere microbial dynamics: Guangxi, China

Xiaoxiao Zhang[1,*], Wei Zhao[1,*], Yuming Lin[1], Bin Shan[2] and Shanshan Yang[1]

[1] Guangxi Key Laboratory of Agro-Environment and Agric-Products Safety, College of Agriculture, Guangxi University, Nanning, China
[2] Guangxi Subtropical Crops Research Institute, Key Laboratory of Quality and Safety Control for Subtropical Fruit and Vegetable, Ministry of Agriculture and Rural Affairs, Nanning, China
* These authors contributed equally to this work.

Corresponding author
Shanshan Yang,
yangshanshan12@126.com

## ABSTRACT

Root-knot nematode disease severely affects the yield and quality of the mandarin variety *Citrus reticulata* Blanco "Orah" in Guangxi, China. Nevertheless, the pathogen and the effects of this disease on microbial communities remain inadequately understood. This study identified the root-knot nematode *Meloidogyne panyuensis* in the rhizosphere of infected Orah using morphological and molecular biological methods. Soil chemical properties indicated that organic matter, total nitrogen (TN), total phosphorus (TP), available phosphorus (AP), total potassium (TK), and available potassium (AK) were significantly higher in the rhizosphere soil of *M. panyuensis*-infected Orah than in that of healthy plants. The relative abundance of the bacteria *Bacillus*, *Sphingomonas*, and *Burkholderia-Caballeronia-Paraburkholderia*, as well as the fungi *Lycoperdon*, *Fusarium*, *Neocosmospora*, *Talaromyces*, and *Tetragoniomyces*, was elevated in the rhizosphere soil of *M. panyuensis*-infected plants. Furthermore, organic matter, TN, available nitrogen (AN), TP, AP, TK, and AK exhibited positive correlations with these bacteria and fungi in the rhizosphere soil of *M. panyuensis*-infected Orah. Potential biocontrol strains, such as *Burkholderia* spp., were identified by comparing the differences in rhizosphere microbial composition between healthy Orah and *M. panyuensis*-infected Orah. Our findings provide a foundation for the early warning and prevention of root-knot nematode disease in Orah.

## INTRODUCTION

Citrus was first cultivated in China, where the planting area and yield are far greater than in any other country. The Guangxi Zhuang Autonomous Region (GZAR) has become one of the main citrus-producing areas in China due to its distinct natural conditions. The

planting area of citrus in Guangxi is stable at approximately $53.3 \times 10^5$ hectares, and the yield exceeds 11 million tons annually. Citrus is the leading industry for local farmers. Increases in market demand and prices have led to the expansion of citrus planting areas in recent years; it has increased from 71,400 hectares in 1961 to 3,189,067 hectares in 2023 (*Deng, 2022*).

*Citrus reticulata* Blanco "Orah" is a hybrid variety bred by Spiegel-Roy and Vardi (*Vardi, Levin & Carmi, 2008*; *Jiang & Cao, 2011*; *He et al., 2022*) that exhibits high yield, good flavor, and superior quality, making it the fastest growing late-maturing citrus variety in China (*Jiang & Cao, 2011*; *Qiu et al., 2022*). The Wuming district of Nanning (a city in GZAR) has vigorously developed its Orah industry. In 2022, the planting area of Orah in Wuming reached 30,700 hectares, with a yield of 1.5 million tons, and the annual output value exceeded \$1.4 billion. This region is now the largest Orah-producing area in China.

The expansion of cultivated areas has increased the occurrence of citrus disease (especially root nematode disease), which restricts the yield and quality of citrus. Citrus root-knot nematodes primarily infect new citrus roots, enlarging new root tips or forming root knots of varying sizes that cannot elongate normally (*He et al., 2020*). Upon infection, the leaves of the plant gradually yellow and fall off, while the top twigs wilt. There are 14 different types of pathogenic root-knot nematodes that can harm citrus, and different regions harbor different species, such as *Meloidogyne incognita* and *M. javanica* in Jiangxi Province and *M. panyuensis* in Sichuan Province (*He et al., 2020*). Root-knot nematode diseases have rapidly spread with the adjustment of cultivation mode and the development of mechanized production, and their occurrence has increased annually. This situation is complex and ever-changing, often involving multiple diseases and nematodes. Identifying pathogenic nematodes and developing targeted prevention and control measures are needed to ensure the stable development of the Orah industry.

Microorganisms in the soil rhizosphere form symbiotic relationships with the host, and the soil rhizosphere microbiome is important for plant health (*Tsang et al., 2020*). Abiotic and biotic factors cause dynamic changes in soil rhizosphere microbial communities. These changes in microbial populations induce variability in soil chemical properties, which can influence plant health (*Ma et al., 2022*). *Pseudomonas fluorescens* 2-79RN$_{10}$ protects wheat against take-all disease, and the biocontrol activity of *P. fluorescens* 2-79RN$_{10}$ is enhanced by the presence of Zn and organic matter (*Ownley, Duffy & Weller, 2003*). Moreover, soil chemical properties influence the soil microbial community and can contribute to soil-borne diseases (*Li et al., 2022*). For example, soil pH and the C to N ratio can affect the microbial community in the rhizosphere and further influence plant health (*Hogberg, Hogberg & Myrold, 2007*; *Janvier et al., 2007*). *Phellinus noxius* infection impacts archaea and fungi in the rhizosphere microbiome (*Tsang et al., 2020*), and the native microbial flora in wilted soil are highly suppressed (*Joshi et al., 2021*). Additionally, the composition of microbial communities in rhizosphere soil can differ significantly between healthy and diseased plants. Healthy watermelons exhibit the lowest abundance of *Fusarium oxysporum* (*Meng et al., 2019*). However, to our knowledge, no studies have focused on the citrus rhizosphere microbiome in relation to root-knot nematode infection.

Plant growth promoting rhizobacteria (PGPR) are important components of the plant rhizosphere soil microbial community and represent a major biocontrol microbial resource (El-Saadony et al., 2022). PGPR maintain the microbial community structure; they are a group of microbes with this community that possess plant growth-promoting traits, such as mineral solubillization efficiecny, siderophore production for iron chelation, or the production of bioactive secondary metabolites to control the growth of various pathogens. They can effectively reduce damage to plants caused by environmental stress, maintain the stability of the soil microbial community, and significantly antagonize pathogen growth (Zhang et al., 2020). Many independent studies have identified Proteobacteria as dominant members of the rhizosphere microbiota, including bacteria from the Pseudomonadaceae or Burkholderiaceae family (Philippot et al., 2013). Since root-knot nematodes can cause significant harm, preventing and controlling root-knot nematode disease is crucial to increasing yield, reducing environmental pollution, and maintaining the sustainable development of Orah agriculture.

In this study, the pathogen associated with Orah root-knot nematode disease was identified in Guangxi. Similarly, the microbial community and the chemical characteristics of the soil in the rhizosphere of diseased and healthy plants were characterized. This information will aid in designing integrated management strategies for the disease.

## MATERIALS AND METHODS

### Sample collection

About 6.7 hectares of Orah (*Citrus reticulata* Blanco) orchard with trees aged 3 years and managed under unified agronomic conditions, was investigated in June 2022 in Wuming District, Nanning City, GZAR (22°58′45.92″N, 108°11′51.24″E) (Huang et al., 2022). Samples of Orah roots and rhizosphere soil were collected from two adjacent orchards with trees at different stages of growth: one healthy and robust and the other short and yellow. Three rows were randomly selected in the healthy or root-knot-infected Orah, and five plants in each row were mixed into one sample. After removing the surface soil, the rhizosphere soil samples were collected. The soil samples were stored in a cooler and transported to the laboratory. The rhizosphere soil was divided into two parts: (1) preserved at −80 °C to determine the rhizosphere soil microbial communities; and (2) air-dried to determine soil chemical properties. The healthy Orah samples were labeled as CRH, while the root-knot-infected Orah samples were labeled as CRN. Nematode egg samples were extracted from the root knots of Orah, purified, and inoculated into the roots of disease-susceptible water spinach (*Ipomoea aquatica* Forsk) in sterilized soil for propagation. Second-stage juveniles (J2s) were collected from the hatching eggs. Simultaneously, the fruits of healthy and nematode-infected Orah were collected to evaluate their mass and diameter.

### Nematode extraction and identification

Female adults were selected from Orah root-knot tissue under an anatomical microscope, and a slide consisting of a 45% lactic acid solution was used to make an impression of the perineal pattern. The tail end of the nematode was cut with a scalpel, and the eggs and

other adhesions attached to the inside were carefully removed and slightly modified with a brush, leaving only the perineal pattern part. The perineal pattern was transferred to a glass slide with one drop of pure glycerol, covered with a cover glass, and photographed under an optical microscope (Yang et al., 2021).

One single clean J2 nematode was placed into a 500 μL centrifuge tube, and 8 μL ddH2O and 1 μL 10× PCR buffer were added. The tube was placed in liquid nitrogen for 1 min, then heated to 85 °C for 2 min, and this process was repeated 7–8 times. Then, 1 μL 1 mg/mL proteinase K was added, followed by incubation at 56 °C for 1 h and 95 °C for 10 min. This nematode genomic DNA was directly used for PCR or stored at −20 °C. The rDNA-ITS sequence of the root-knot nematodes was amplified using the universal primers: V5367/26S (5′-TTGATTACGTCCCTGCCCTTT-3′; 5′-TTTCACTCGCCGTT ACTAAGG-3′) (Vrain, 1993) using the extracted DNA as a template. The obtained rDNA-ITS sequences were compared to other sequences using BLAST (NCBI), and molecular phylogenetic analysis was constructed using the maximum likelihood method of the MEGA7.0 software, with the number of replicates for the tree-building step set at 1,000 (bootstrap replicates = 1,000). The PCR reaction system was performed as follows: 2 μL DNA template, 2 μL of each primer, 25 μL of 2×Rapid Taq Master Mix, and 21 μL ddH2O. The PCR reaction procedure included initial denaturation at 95 °C for 3 min, followed by 35 cycles of 95 °C for 30 s, 55 °C for 30 s, and 72 °C for 30 s, and a final extension step at 72 °C for 5 min, followed by storage at 4 °C. The PCR products were examined and sequenced. In addition, specific primers were used to detect nematodes: Mp-F/Mp-R (5′-GTTTTCGGCCCGCAACATGT-3′ and 5′-CACCGCCTTGCGTAAACTCC-3′). The PCR reaction system was described above. The PCR reaction procedure included initial denaturation at 95 °C for 3 min followed by 30 cycles of 95 °C for 30 s, 63 °C for 30 s, 72 °C for 45 s, and a final extension at 72 °C for 5 min, followed by storage at 4 °C. Amplification products were examined by 1% agarose gel electrophoresis for the appearance of a single band of the expected size of 409 bp (He et al., 2020).

Healthy Orah seedlings were purchased from the Guangxi Subtropical Crops Research Institute and transplanted into a pot containing sandy loam soil for growth in a greenhouse at room temperature under natural light. After survival, 1,000 J2 nematodes were inoculated into the roots of 10 Orah seedlings, while another 10 Orah seedlings without nematodes were used as a negative control. The root soil was removed after 3 months to observe the damage caused by the nematodes to the roots.

## Soil chemical properties

The chemical properties including soil pH, organic matter, total nitrogen (TN), available nitrogen (AN), total phosphorus (TP), available phosphorus (AP), total potassium (TK), and available potassium (AK) were determined as described previously (Bao, 2000). Soil pH was determined using a composite glass electrode meter with a soil: water ratio of 1:2.5. Soil TN was determined using the semi-micro Kjeldahl method; soil AN was determined using a Kjeldahl nitrogen meter; soil TP and AP were determined using the molybdenum-antimony anti-colorimetric method; and soil TK and AK were determined using the flame photometric method.

## Soil DNA isolation and PCR conditions

Total DNA was extracted from each rhizosphere soil sample (0.4 g) using the MoBio PowerSoil kit according to the manufacturer's instructions. The purity and concentration of the extracted DNA were quantified using a Nanodrop spectrophotometer (ND-2000) and on 0.5% agarose gels. Bacterial communities were determined using the universal forward primer (338F:5′-ACTCCTACGGGAGGCAGCAG-3′) and reverse primer (806R:5′-GGACTACHVGGGTWTCTAAT-3′) to amplify the V3–V4 region of the 16S rRNA gene (*Guo et al., 2019*). Fungal communities were examined using the *ITS1* gene, amplified using the universal forward primer (ITS1F:5′- CTTGGTCATTTAGAGGAA GTAA-3′) and reverse primer (ITS2R:5′- GCTGCGTTCTTCATCGATGC-3′) (*Li et al., 2020*). The PCR reaction was performed in a 20 μL volume containing 5×FastPfu Buffer 4 μL, 2.5 mM dNTPs 2 μL, each primer 0.8 μL, FastPfu Polymerase 0.4 μL, bovine serum albumin (BSA) 0.2 μL, and soil genomic DNA 10 ng. Amplifications were performed using an initial denaturation step of 95 °C for 3 min, followed by 30 cycles of denaturation at 95 °C for 30 s, annealing at 50 °C for 30 s, and extension at 72 °C for 45 s; with a final extension step at 72 °C for 7 min, then stored at 4 °C. The PCR products were purified with a PCR Clean-Up$^{TM}$ kit (MO BioLabs, Carlsbad, CA, USA) and sent to the Majorbio Company (Shanghai, China) for sequencing using an Illumina MiSeq PE300 (*Edgar, 2013*).

## High-throughput sequencing analysis

The raw sequenced reads were quality-controlled (QC) using fastp software (https://github.com/OpenGene/fastp, version 0.20.0) (*Chen et al., 2018*), and spliced using FLASH software (http://www.cbcb.umd.edu/software/flash, version 1.2.7) (*Magoc & Salzberg, 2011*). Quality control involved: filtering bases with quality values below 20 at the end of the reads, setting a window of 50 bp, truncating back-end bases from the window if the average quality value within the window was below 20, filtering reads below 50 bp after QC, and removing reads containing N bases. Pairs of reads were spliced (merged) into one sequence with a minimum overlap length of 10 bp according to the overlap relationship between paired end double-end sequencing reads. The maximum mismatch ratio in the overlap region of the spliced sequence was 0.2, and the non-conforming sequences were screened. The samples were distinguished according to the barcode and primers at the beginning and end of the sequence, and the sequence orientation was adjusted. The number of mismatches in the barcode was 0, and the maximum number of primer mismatches was 2. The data were used for subsequent bioinformatics analyses.

All data analyses were performed using the Meguiar BioCloud platform (https://cloud.majorbio.com). Alpha diversity Coverage, and Chao 1, Simpson, Shannon, and Sobs indices were calculated using mothur software (http://www.mothur.org/wiki/Calculators), and the Wilcoxon rank-sum test was used to analyze group differences in Alpha diversity and to complete the dilution curve analysis; the similarity of microbial community structure between samples was examined using principal coordinate analysis (PCoA) based on the Bray-Curtis distance algorithm; the analysis of microbial community composition was performed with python software (https://www.python.org); the Wilcoxon rank-sum test, two-tailed test, bootstrap algorithm for microbial community

inter-group variation and Bray-Curtis distance algorithm for correlation with environmental factors were performed using R-3.3.1 software (*R Core Team, 2016*).

## Statistical analysis

IBM SPSS Statistics 20 was used to analyze significant differences at the $p < 0.05$ level according to the Duncan multiple range test in the growth indicators (the fruit mass and diameter of Orah), soil chemical properties, and ecological indicators (Coverage, Chao 1, Simpson, and Shannon indices). The redundancy analysis (RDA) through the analysis software R-3.3.1 (vegan) was relied upon to examine the relationships among the soil chemical properties and microbial communities.

# RESULTS

## Root-knot nematode infection decreased Orah yield

In June, the growth of many Orah infected by root-knot nematode, was weak. Compared with healthy roots, the roots infected by nematodes had root knots of different sizes, and the root disks were together to form fibrous roots, which were messy (Figs. 1A, 1B). The mass and diameter of fruits were significantly lower in root-knot nematode infected Orah than in healthy Orah (Fig. 1C). The average fruit diameter of healthy Orah was 3.86 cm, while the average fruit diameter of root-knot nematode infected Orah was 2.95 cm. The average fruit diameter of root-knot nematode infected Orah was reduced by 23.58% compared to that of healthy Orah (Fig. 1D). The average fruit mass of root-knot nematode infected Orah was 7.49 g, while the average fruit mass of healthy Orah was 19.48 g. The average fruit mass of root-knot nematode infected Orah was reduced by 61.55% compared to that of healthy Orah (Fig. 1E).

## Identification of root-knot nematode infecting Orah

There was slight variation among individuals in the female perineal pattern population, but the degree of variation among populations was similar. Microscopic observation of root-knot nematode populations isolated from Orah showed that the root-knot nematode was *M. panyuensis*: the female was white, spherical to pear-shaped, with an obvious neck, and the neck ring was clear. The back ring was not obvious, and the excretory orifice was located at the median bulb. The pin was developed, and the basal knob was thick. The perineal pattern was oval, the line was smooth, the back arch was low, and there was no obvious side line (Fig. 2A). The morphology of the nematode was consistent with that previously reported (*He et al., 2020*).

The length of the obtained rDNA-ITS sequence was 869–870 bp (GenBank accession numbers OR135523–OR135524), Blast results confirmed that those sequences were 97.59–99.08% identical to those of *M. panyuensis* sp. n. from *Arachis hypogaea* L. in Guangdong, China (AY394719.1) (*Liao et al., 2005*). The phylogenetic tree was constructed based on the rDNA-ITS sequences, and the results showed that *M. panyuensis* from Guangxi was clustered with *M. panyuensis* sp. n. from Guangdong (AY394719.1) (*Liao et al., 2005*) within a group at a value of 100% (Fig. 2B). A single 409 bp specific

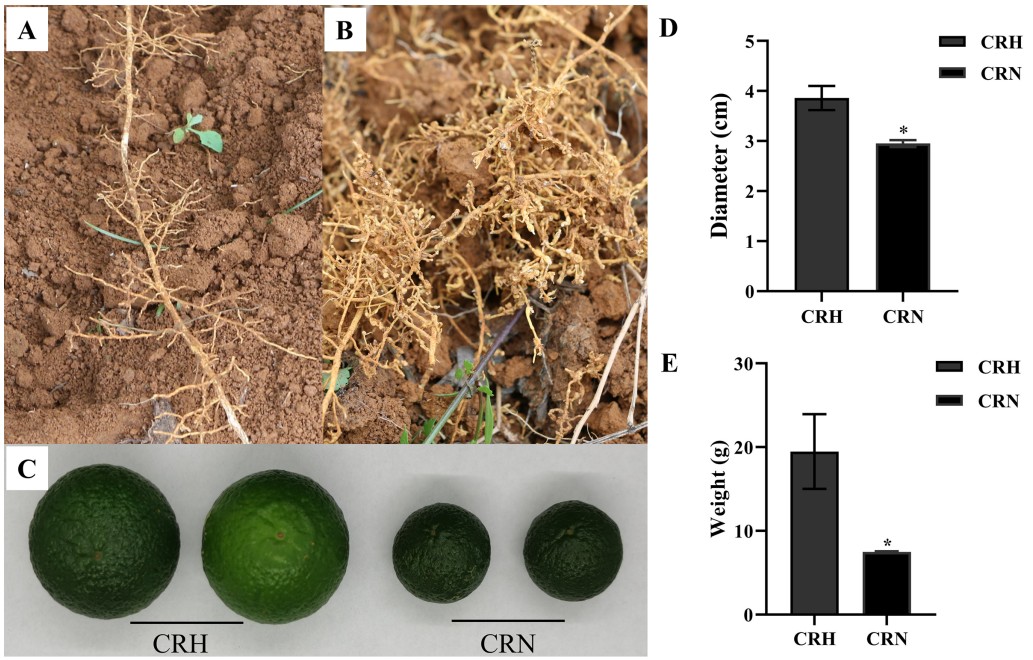

**Figure 1 Damage symptom of root-knot nematodes on Orah and its effect on yield.** (A) Roots of healthy Orah. (B) Roots of root-knot nematodes infected Orah. (C) The fruit appearance of the healthy Orah (CRH) and the root-knot nematodes infected Orah (CRN). (D) The diameter of fruits. (E) The mass of fruits. * represents a significant difference ($p < 0.05$).

fragment was obtained through PCR amplification using the DNA of root-knot nematodes as a template and the specific primers Mp-F/Mp-R of *M. panyuensis* (Fig. 2C).

Cultured J2s were inoculated into healthy Orah seedlings with good growth and cultured at room temperature for 3 months. The inoculated roots were found to produce root knots (Figs. 2D, 2E).

## Soil chemical properties

Infection with *M. panyuensis* did not significantly affect the pH but had a great influence on organic matter, TN, AN, TP, AP, TK, and AK (Table 1). The organic matter, AP, AK, TN, TP, and TK significantly increased in *M. panyuensis*-infected Orah rhizosphere soil by 58.87%, 14.39%, 521.39%, 37.14%, 52.31%, and 30.34%, respectively compared with those in healthy Orah rhizosphere soil.

## General analyses of the sequencing data

The V3–V4 region of bacterial 16S rDNA and fungal ITS high-throughput sequencing results from Orah rhizosphere soil samples were analyzed, yielding 348,051 and 382,559 optimized sequences, respectively. Furthermore, 3,509 and 1,085 amplicon sequence variant (ASV) sets were generated using the clustering method. The cluster analysis of the ASV sets covered over 99% of the Orah rhizosphere soil microorganisms, indicating that the sequencing results reflected changes in the citrus rhizosphere soil microbial community. The dilution curves of the Shannon and Sobs indices showed a gentle slope

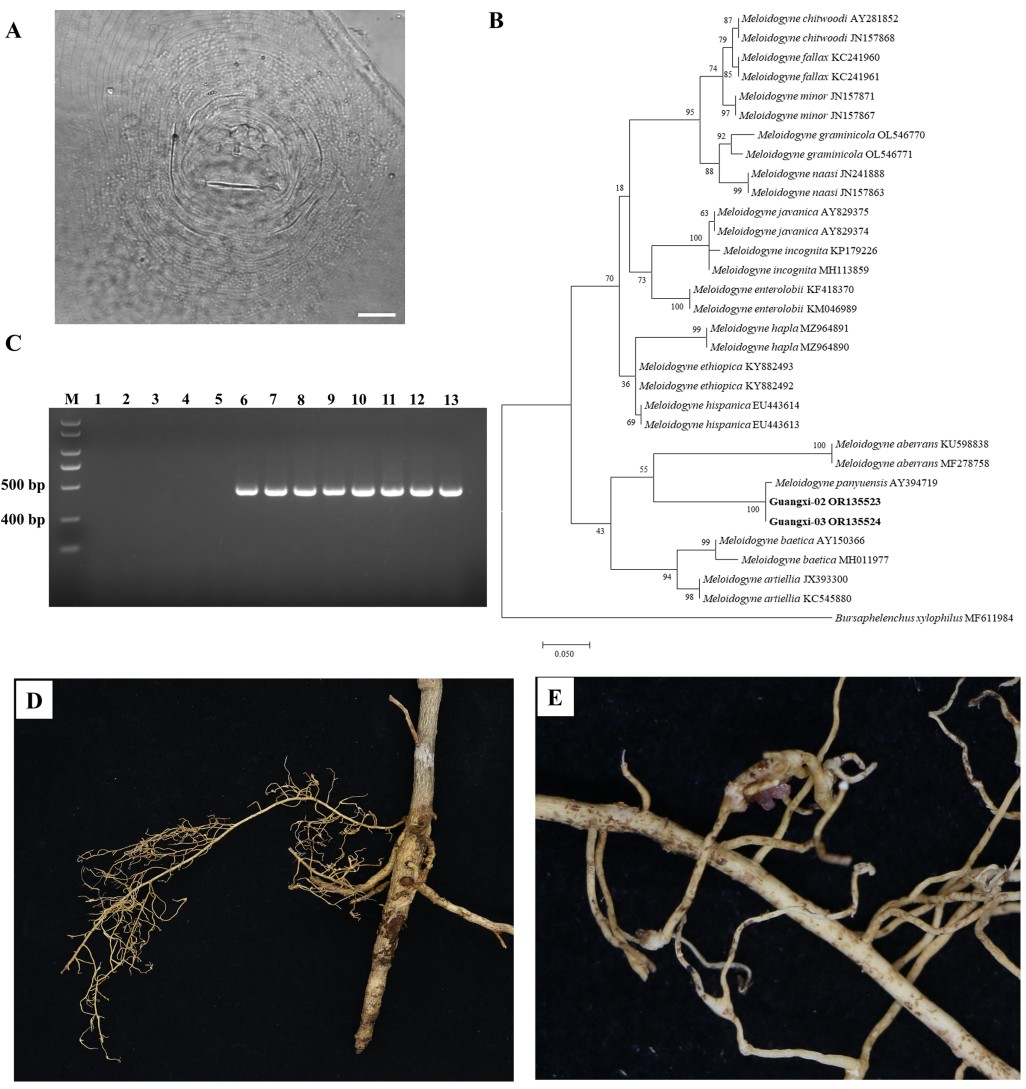

**Figure 2 Identification of rot-knot nematode infecting Orah.** (A) Perineal pattern of nematode female isolated from infected Orah root. Scale bar: 20 μm. (B) Phylogenetic tree based on rDNA-ITS of *Meloidogyne* spp. (C) Specific amplification using the primers Mp-F/Mp-R in *M. panyuensis* isolated from infected Orah. M: DL 2000 Plus; 1–5: negative control (1. Water; 2–3. *M. incognita*; 4–5. *M. enterolobii*.); 6–13: *M. panyuensis*. (D–E) The disease symptoms on Orah caused by *M. panyuensis*.

with increasing sequencing depth (Fig. S1), suggesting that the sequencing depth was sufficient to capture the vast majority of soil bacteria and fungi. Additionally, the ecological index and coverageindicated that the current sequencing depth was adequate to cover more than 99% of soil bacteria and fungi (Table 2).

## Soil bacterial and fungal diversity

The diversity of bacterial and fungal communities in healthy Orah rhizosphere soils was compared with those in *M. panyuensis*-infected soils. The community richness index (Chao1 index) and community diversity (Shannon index) of bacteria in *M. panyuensis-*

**Table 1 Soil chemical properties of the healthy and *M. panyuensis*-infected Orah rhizosphere soil.**

| Number | Samples | pH | Organic matter (g/kg) | AN (mg/kg) | AP (mg/kg) | AK (mg/kg) | TN (g/kg) | TP (g/kg) | TK (g/kg) |
|---|---|---|---|---|---|---|---|---|---|
| 1 | CRH | 5.02 ± 0.94 a | 23.10 ± 4.70 a | 123.95 ± 25.55 a | 44.56 ± 28.04 a | 215.42 ± 68.84 a | 1.40 ± 0.13 a | 1.30 ± 0.07 a | 2.34 ± 0.04 a |
| 2 | CRN | 5.06 ± 0.20 a | 36.70 ± 1.21 b | 141.79 ± 3.66 a | 276.89 ± 128.40 b | 430.40 ± 26.75 b | 1.92 ± 0.06 b | 1.98 ± 0.25 b | 3.05 ± 0.20 b |

Note:

The data are presented as means ± standard errors, and different lowercase letters indicate significant differences at the 0.05 level.

**Table 2 Richness and diversity estimation of the 16S and ITS1 sequencing libraries in healthy and *M. panyuensis*-infected Orah rhizosphere soil.**

| Category | Group | Coverage | Chao 1 | Simpson | Shannon |
|---|---|---|---|---|---|
| Bacteria | CRH | 0.9996 | 840 ± 77 a | 0.003 ± 0.0003 a | 6.26 ± 0.07 a |
| | CRN | 0.9992 | 937 ± 133 a | 0.0025 ± 0.0003 a | 6.39 ± 0.14 a |
| Fungi | CRH | 1 | 370 ± 75 a | 0.0469 ± 0.0093 a | 3.98 ± 0.1 a |
| | CRN | 1 | 262 ± 35 a | 0.0761 ± 0.02 a | 3.5 ± 0.29 a |

Note:

The data are presented as means ± standard errors, and different lowercase letters indicate significant differences at the 0.05 level.

infected Orah soil increased, whereas community diversity (Simpson index) decreased. Meanwhile, the Chao1 and Shannon indices of fungi in *M. panyuensis*-infected Orah soil decreased, whereas the Simpson index increased. However, there was no significant difference between the healthy and *M. panyuensis*-infected Orah rhizosphere soils (Table 2). The *M. panyuensis*-infected Orah rhizosphere soil bacterial community and the healthy one were separated according to PCoA; the combined horizontal and vertical axes in the figure explained 70.6%. The analysis of similarity (ANOSIM) discriminant coefficient R was 0.1852 (Fig. 3A). However, the result was not significantly different ($p > 0.05$). The *M. panyuensis*-infected Orah rhizosphere soil fungal community partly overlapped with the healthy Orah soil bacterial community. The combined horizontal and vertical axes explained 68.6% of the variance, and the ANOSIM discriminant coefficient R was 0.3704, but the difference was no significant ($p > 0.05$, Fig. 3B). These results showed that there was no significant difference in rhizosphere soil community diversity between healthy and *M. panyuensis*-infected Orah rhizosphere soils.

## Soil bacterial and fungal taxonomic composition

The *M. panyuensis*-infected and healthy Orah rhizosphere soils had the same bacterial community composition, with 229 orders detected. Among them, the relative abundances of Rhizobiales, Burkholderiales, Acidobacteriales, Bacillales, Sphingobacteriales, Bryobacterales, Vicinamibacterales, Frankiales, Chitinophagales, Ktedonobacterales, Gaiellales, and Xanthomonadales were all above 2%, and represented the dominant bacteria in Orah rhizosphere soil. The abundances of the remaining 217 orders were below 2%. The abundance of Burkholderiales in *M. panyuensis*-infected Orah soil (63%) was significantly higher than the 37% in healthy Orah soil (Fig. 4A). In total, 609 genera were identified. Among them, *Bacillus, Bryobacter, Sphingomonas, Acidothermus, Burkholderia-*
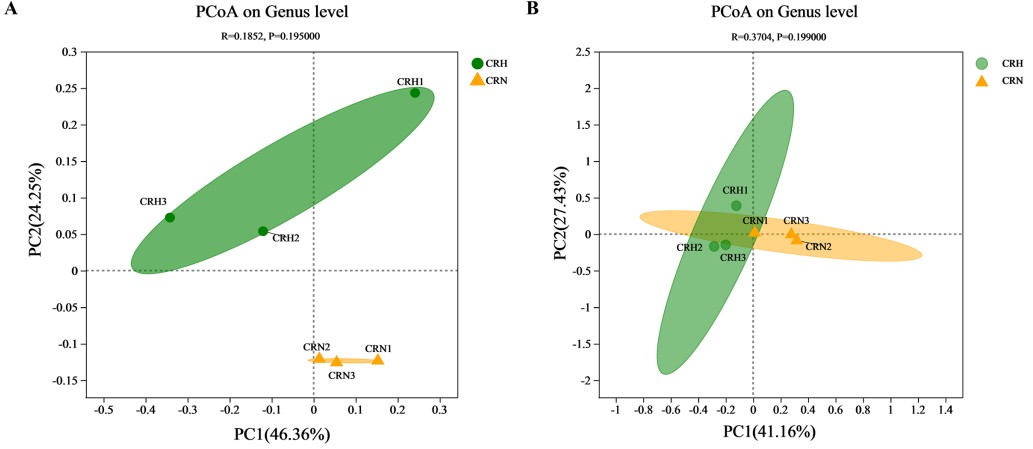

**Figure 3** Principal coordinate analysis (PCoA) of bacterial (A) and fungal (B) communities based on Bray-Curtis distances.

*Caballeronia-Paraburkholderia*, *norank-f-norank-o-Acidobacteriales*, *norank-f-norank-o-Vicinamibacterales*, *norank-f-norank-o-Gaiellales* and *norank-f-Xanthobacteraceae* accounted for over 2%, and were the dominant bacteria in Orah rhizosphere soil. The abundances of the other 600 genera were below 2%. The abundance of *Burkholderia-Caballeronia-Paraburkholderia* in *M. panyuensis*-infected Orah soil was 84% (Fig. 4B). Which was significantly higher than the 16% in healthy Orah soil. These results indicate that *Burkholderia-Caballeronia-Paraburkholderia* were closely related to the occurrence of nematode disease.

*M. panyuensis*-infected and healthy Orah rhizosphere soils had the same fungal community composition, with 11 phyla detected. Among these, the relative abundances of Ascomycota, Basidiomycota, unclassified_k_Fungi, Rozellomycota, and Mortierellomycota were above 1%, and represented the dominant phyla in the rhizosphere soil. The abundances of the other six phyla were below 1%. The abundance of Basidiomycota in *M. panyuensis*-infected Orah rhizosphere soil was 81%; this was significantly higher than the 19% in healthy Orah rhizosphere soil (Fig. 4C). A total of 275 genera were identified. Among these, *Lycoperdon*, *Roussoella*, *Fusarium*, *Neocosmospora*, *Penicillium*, *Chaetomium*, *Talaromyces*, *Acrocalymma*, *unclassified_k_Fungi*, and *Tetragoniomyces* accounted for over 3%, whereas the remaining 265 genera accounted for below 3%. The abundance of *Lycoperdon*in *M. panyuensis*-infected Orah rhizosphere soil was 99%; this was significantly higher than the 1% in healthy Orah rhizosphere soil (Fig. 4D). This indicated that there was a connection between *Lycoperdon* and the occurrence of nematode disease.

## Comparative analysis between groups of healthy and *M. panyuensis*-infected Orah soil microbial communities

Comparative analysis of bacteria in Orah rhizosphere soil samples indicated that the relative abundance of *Burkholderia-Caballeronia-Paraburkholderia* in *M. panyuensis*-infected Orah soil was 4.71%, which represents a 5.44-fold increase compared to healthy

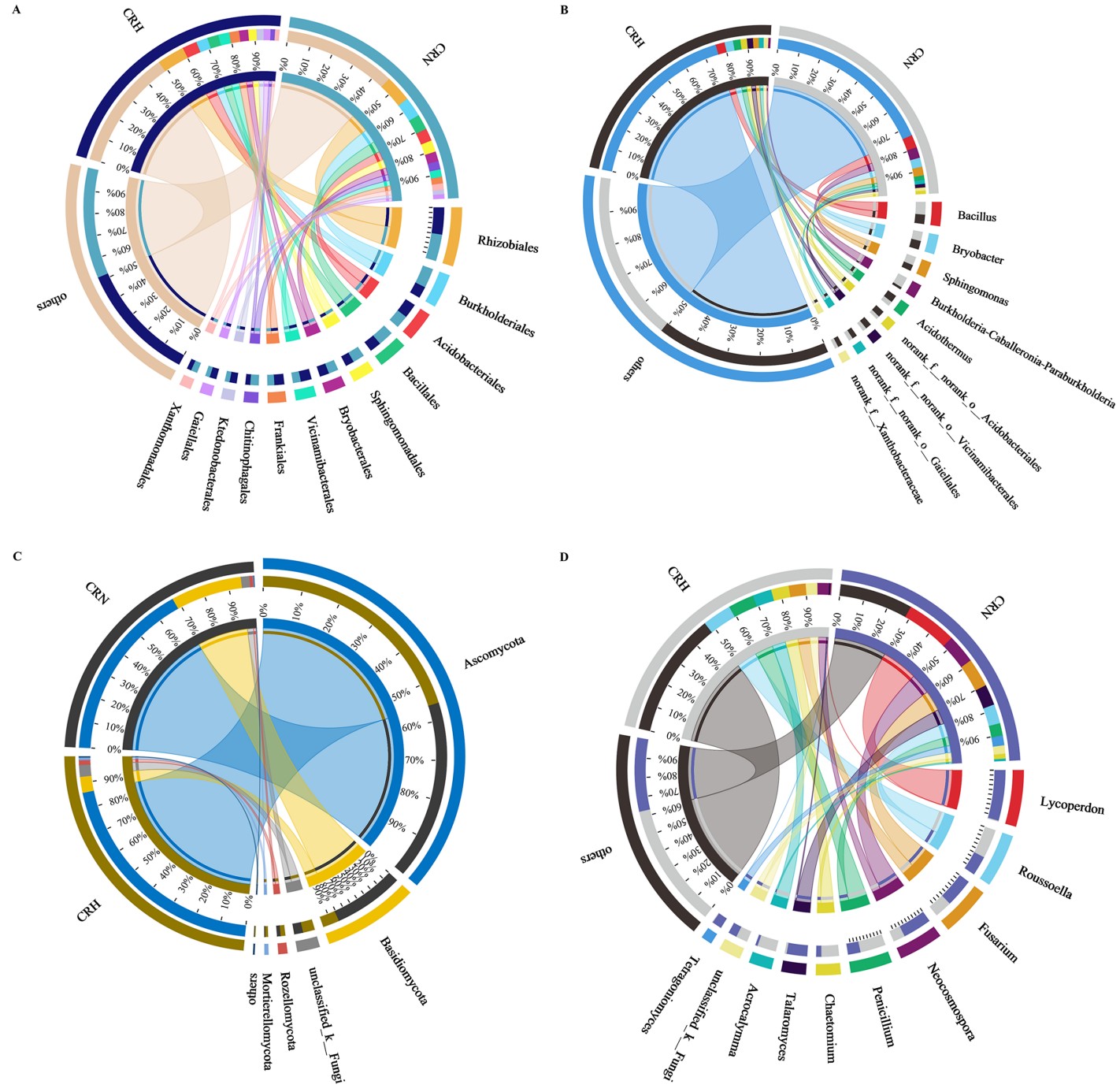

**Figure 4 Relationship between species and samples in rhizosphere soil of healthy and *M. panyuensis*-infected.** (A) (Order level) and (B) (genus level) were the bacterial community; (C) (phylum level) and (D) (genus level) were the bacterial community.

Orah rhizosphere soil groups (0.87%). The relative abundance of *Bacillus* in *M. panyuensis*-infected Orah rhizosphere soil was 5.53%, reflecting a 1.40-fold increase over the healthy Orah rhizosphere soil group (3.96%). The relative abundance of *Sphingomonas* in *M. panyuensis*-infected Orah rhizosphere soil was 3.75%, representing a 1.50-fold

**A**

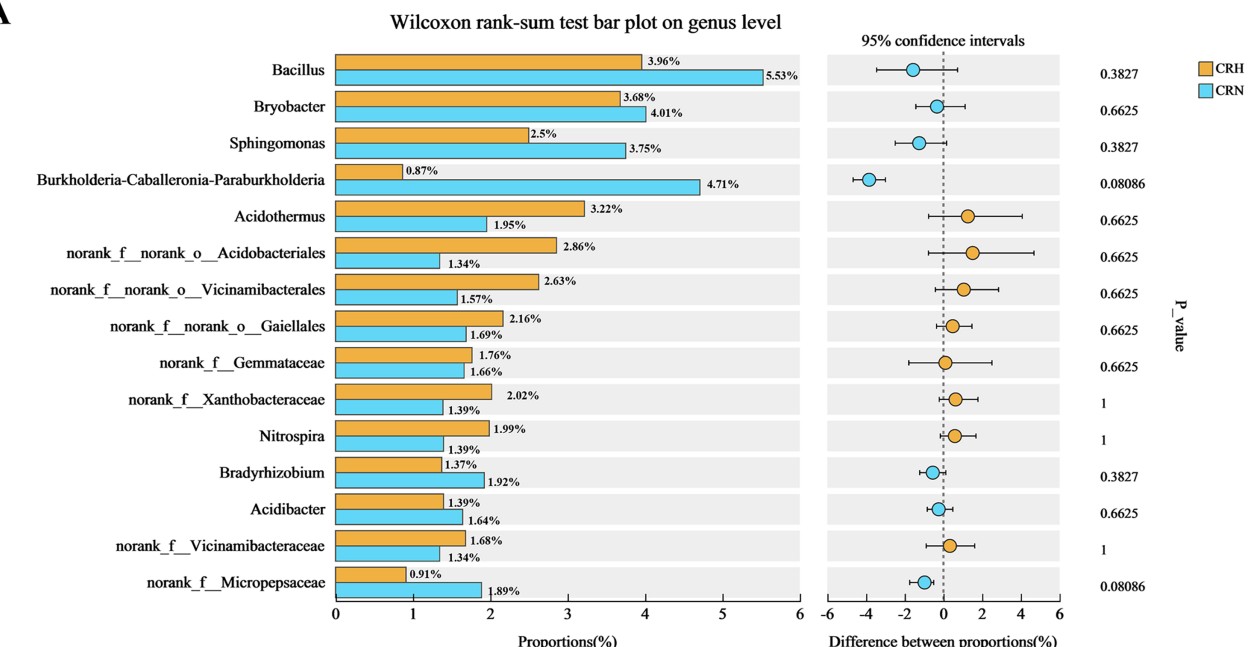

**B**

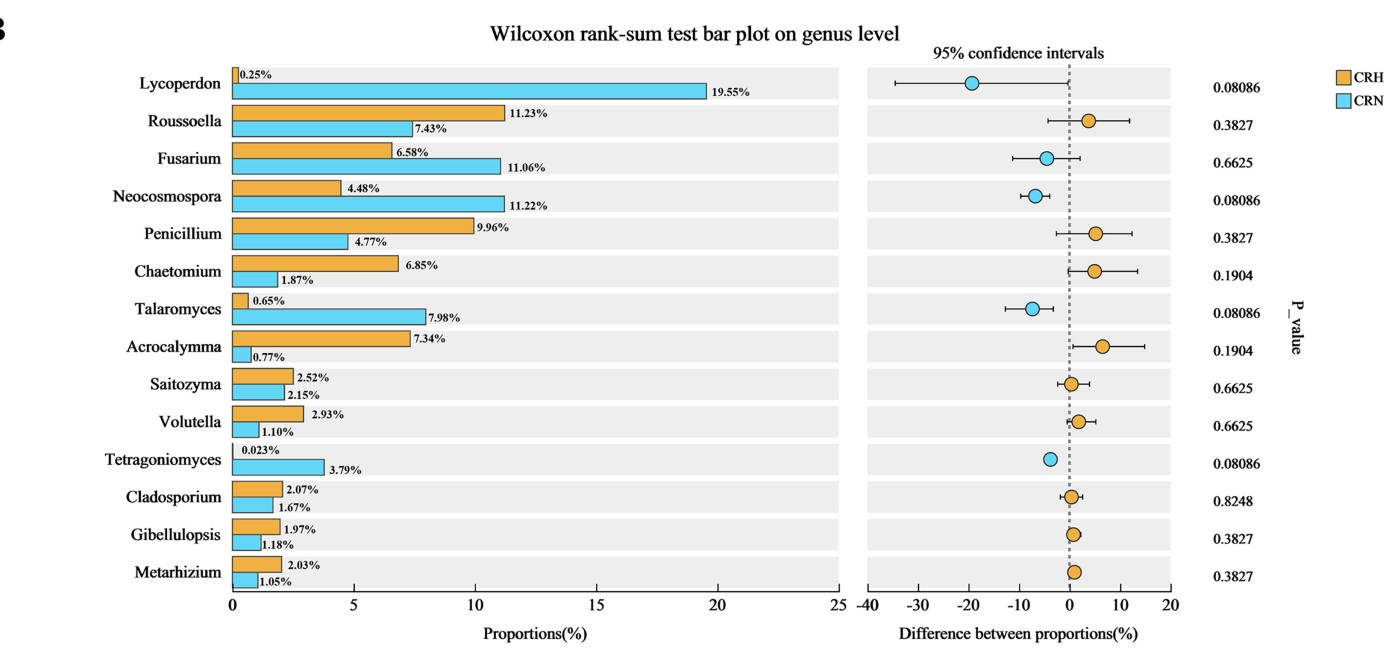

**Figure 5 Comparative analysis of bacterial (A) and fungal (B) community in rhizosphere soil of healthy and *M. panyuensis*-infected.**

increase compared to healthy Orah rhizosphere soil groups (2.50%) (Fig. 5A). These results further indicated a relationship between *Burkholderia-Caballeronia-Paraburkholderia, Bacillus, Sphingomonas*, and nematode diseases.

Similarly, the comparative analysis of fungi in Orah rhizosphere soil samples revealed that the relative abundance of *Lycoperdon* in *M. panyuensis*-infected Orah rhizosphere soil was 19.55%, a 77.01-fold increase over healthy Orah rhizosphere soil groups (0.2506%).

The relative abundance of *Fusarium* in *M. panyuensis*-infected Orah rhizosphere soil was 11.06%, which was 1.68-times greater than that in healthy Orah rhizosphere soil groups (6.58%). The relative abundance of *Neocosmospora* in *M. panyuensis*-infected Orah soil was 11.22%, indicating a 2.50-fold increase compared to healthy Orah rhizosphere soil groups (4.48%). The relative abundance of *Talaromyces* in *M. panyuensis*-infected Orah rhizosphere soil was 7.98%, showing a 12.28-fold increase over the healthy Orah soil group (0.65%). The relative abundance of *Tetragoniomyces* in *M. panyuensis*-infected Orah rhizosphere soil was 3.79%, which represents a 189.50-fold increase compared to the healthy Orah rhizosphere soil group (0.02%) (Fig. 5B). These findings further suggested a relationship between *Lycoperdon*, *Fusarium*, *Neocosmospora*, *Talaromyces*, *Tetragoniomyces*, and nematode disease.

## Correlation analysis between healthy and *M. panyuensis*-infected Orah soil microbial communities and environmental factors

RDA showed that soil organic matter and available N, P, and K were significantly and positively correlated with the bacterial community in *M. panyuensis*-infected Orah rhizosphere soil. The explanatory degrees of the horizontal and vertical coordinates were 46.02% and 28.01%, respectively. Soil organic matter and available K showed strong correlation coefficients (r2) of 0.9982 and 0.9693, respectively (Fig. 6A). Total N, P, and K were significantly positively correlated with the bacterial community in *M. panyuensis*-infected Orah rhizosphere soil. The interpretations of the horizontal and vertical coordinates were 39.05% and 27.83%, respectively. Total N and K showed a strong positive correlation, with correlation coefficients ($r^2$) of 0.9342 and 0.9055, respectively (Fig. 6B). In addition, four of the ten genera with the highest relative abundance were positively correlated with the bacterial community in *M. panyuensis* infected Orah rhizosphere soil, including *Burkholderia-Caballeronia-Paraburkholderia* (Figs. 6A, 6B). These results add further evidence that *Burkholderia-Caballeronia-Paraburkholderia* were related to nematode infection.

The soil organic matter and available N, P, and K were significantly positively correlated with the fungal community in *M. panyuensis*-infected Orah rhizosphere soil. The horizontal and vertical coordinates were 58.18% and 20.09%, respectively. Soil organic matter showed a strong positive correlation with a correlation coefficient ($r^2$) of 0.9116 (Fig. 6C). Total N, P, and K were significantly positively correlated with the fungal community in *M. panyuensis*-infected Orah rhizosphere soil. The interpretations of the horizontal and vertical coordinates were 47.58% and 20.74%, respectively. Among these, total P and total K showed strong positive correlations, with correlation coefficients ($r^2$) of 0.9031 and 0.8856, respectively (Fig. 6D). In addition, four of the ten genera with the highest relative abundance were positively correlated with the fungal community in the *M. panyuensis*-infected Orah rhizosphere soil, including *Lycoperdon* (Figs. 6C, 6D). These results provided further evidence that *Lycoperdon* was related to nematode infection.

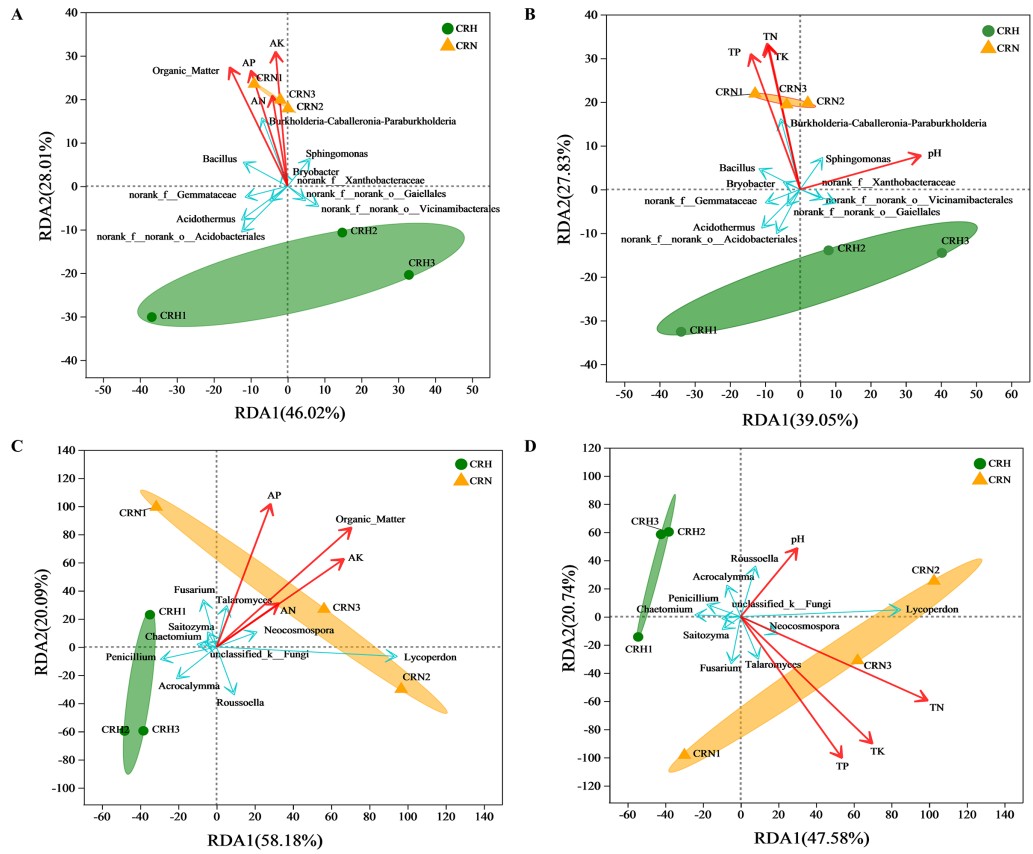

**Figure 6 Redudancy analysis (RDA) between healthy and *M. panyuensis*-infected Orah rhizosphere soil microbial communities and environmental factors.** (A & B) Bacterial community; (C & D) fungal community.

## DISCUSSION

In recent years, Wuming District of Nanning City has actively developed the Orah industry, with the production base driving product processing and promoting the development of the entire Orah industrial chain. In 2022, Wuming became the largest Orah production area in China, but root knot nematode diseases threatened the Orah yield and quality. Identifying plant nematode pathogens is a prerequisite for researching nematode diseases. This study identified the root-knot nematode collected from Orah in Wuming. Morphological and molecular analyses revealed that the pathogenic nematode was *M. panyuensis*. *M. panyuensis* sp. n. was first isolated from peanuts in Guangzhou Province (*Liao et al., 2005*). This nematode has also beendetected in guava and pepper in Hainan, and citrus in Hunan (*Rui et al., 2005*; *Wang, Li & Peng, 2007*). This is the first study demonstrating that *M. panyuensis* harms Orah in GZAR.

Rhizosphere microorganisms that constitute the core of plant rhizosphere functions play important roles in plant growth and development. They are regarded as the second genome of plants and have gradually become key regulatory areas for promoting green agricultural development (*Zhang, 2020*). Rhizosphere soil microorganisms coordinate plant growth and development by increasing the mobility of nutrients in the soil and
helping plants resist pathogen attacks (*Cook et al., 1995*; *Haney et al., 2015*). This study was the first to examine the effects of *M. panyuensis* on the structure and diversity of bacterial and fungal rhizosphere microbiomes in Orah. A comparison of the rhizosphere soil from root-knot nematode-infected and healthy Orah showed no significant difference in the community diversity of bacteria and fungi. The effect of *M. panyuensis* infection on rhizosphere microorganisms was mainly manifested in the selective abundance of microbial populations.

The bacterial community in rhizosphere soil plays a key role in suppressing soil-borne plant diseases (*Ling et al., 2014*; *Wang et al., 2017*). The bacterial community of Orah infected with *M. panyuensis* was significantly different from that of healthy Orah and corresponded to increased disease severity. This was consistent with previous reports; for example, the soil bacterial community suppresses plant soil-borne diseases in developing soil upon organic fertilization, and contributes to the control of *Fusarium* sp. ACCC 36194 (*Chen et al., 2020*). In this study, the relative abundances of *Acidothermus* and *norank_f_norank_o_Acidobacteriales* were significantly higher in healthy rhizosphere soil, whereas the relative abundances of *Bacillus*, *Sphingomonas*, and *Burkholderia-Caballeronia-Paraburkholderia* were higher in *M. panyuensis*-infected rhizosphere soil. *Acidothermus* can decrease soil pH and significantly increase fertility (*Ding et al., 2020*; *Ma et al., 2009*), while *norank_f_norank_o_Acidobacteriales* can produce antibiotics, enzymes, and organic acids with high degradation characteristic (*Ellis et al., 2003*; *Sang-Hoon, Jong-Ok & Jae-Chang, 2008*). The increase in *Acidothermus* and *norank_f_norank_o_Acidobacteriales* in healthy rhizosphere soil may be beneficial for the growth of Orah and could be used as potential probiotic for root-knot nematode disease in Orah. *Bacillus* has strong stress resistance and can secrete various hydrolytic enzymes such as lipase, protease, and amylase. It has a wide antibacterial spectrum, fast reproduction, low production cost, and high safety and is widely used in agricultural disease prevention and control (*Wang, Wang & Li, 2021*; *Kannan et al., 2022*). The increase in its relative abundance may be related to its resistance to root-knot nematode disease. *Sphingomonas* is an abundant microbial resource that can be used to biodegrade organophosphorus compounds and readily degrades pesticides (*Nneby, Jonsson & Stenstrm, 2010*; *Sharp et al., 2012*). The increase in its relative abundance might be related to the degradation of pesticides used to control nematode diseases. *Burkholderia*, which shows the most significant difference, is widely distributed in soil, water, and plant rhizospheres, and is an important biocontrol and growth-promoting bacterium. Extracellular enzymes produced by this genus can dissolve insoluble phosphorus in soil, promote plant growth, and produce a variety of secondary metabolites that inhibit different fungal diseases (*El-Banna & Winkelmann, 1998*; *Parke & Gurian-Sherman, 2001*; *Wang et al., 2011*; *Kim et al., 2012*; *Gong et al., 2019*). The increase in *Burkholderia* may inhibit the growth of pathogens such as *M. panyuensis*, although this requires further research.

The fungal community in the rhizosphere soil plays an important role in the soil ecological environment. In agricultural production, long-term monoculture and continuous cropping can lead to changes in fungal community diversity. The fungal community of Orah infected with *M. panyuensis* was significantly different from that of

healthy Orah and corresponded to the upregulation of disease severity. This was consistent with previous reports, for example, the abundance of soil-borne disease pathogens *Fusarium* and *Guehomyces* increased, whereas that of nematocidal fungi (*Arthrobotrys*) significantly decreased (*Li & Liu, 2019*). In this study, the relative abundance of *Penicillium*, *Chaetomium*, and *Acrocalymma* was significantly higher in the healthy rhizosphere soil, while that of *Lycoperdon*, *Fusarium*, *Neocosmospora* and *Talaromyces* were higher in *M. panyuensis*-infected rhizosphere soil. Endophytic *Penicillium* can colonize their ecological niches and protect host plants against multiple stresses (*Toghueo & Boyom, 2020*), *Chaetomium* has potential biocontrol ability, and its metabolites contain a variety of chemicals that can improve soil fertility, stimulate plant growth, and enhance the antioxidant capacity of certain tissues to boost disease resistance (*Fu & Zhang, 2012*; *Gao et al., 2006*; *Pothiraj et al., 2021*). Moreover, *Acrocalymma* can suppress soil-borne fungal diseases in cucumbers (*Huang et al., 2020*). The increase in *Penicillium*, *Chaetomium*, and *Acrocalymma* might represent potential probiotic benefit for the growth of Orah. *Fusarium* (*Simes, Diogo & Andrade, 2022*; *Sun et al., 2022*) and *Neocosmospora* (*Gai et al., 2011*; *Sun et al., 2014*) are pathogens that cause numerous diseases. Infestation by root-knot nematodes leads to severe diseases due to their association with other microorganisms. These pathobiomes are referred to *Meloidogyne*-based disease complexes (*Wolfgang et al., 2019*). The increase in the abundance of *Fusarium* and *Neocosmospora* might be closely related to infection by *M. panyuensis*. *Talaromyces* is an important biocontrol fungus that has a hyperparasitic effect on various pathogens, and its chitinase exhibits strong antibacterial activity (*Marois, Fravel & Papavizas, 1984*; *Madi et al., 1997*; *Xian et al., 2012*). In the present study, the increase in *Talaromyces* abundance may be related to the inhibition of *M. panyuensis*. The biocontrol function of *Lycoperdon*, which shows the most significant difference, has not yet been reported, indicating a need for further study.

Environmental variables, like soil elements, are closely related to microbial communities, and they can interact (*Li et al., 2009*, *2023*). The available carbon content mainly affects the functional activity of microorganisms in soil. Available soil nutrients can cause changes in bacterial and archaeal communities, and a high content of soil nitrogen promotes excessive growth of plants to a certain extent (*Philippot et al., 2013*; *Tian et al., 2016*). In this study, the contents of organic matter, TN, AN, TP, AP, TK, and AK in the *M. panyuensis*-infected Orah rhizosphere soil were all higher than those in healthy rhizosphere soil. This finding is consistent with a previous report showing that the TN, AN, TP, AP, TK, and AK of *Fusarium* wilt-infected watermelon rhizosphere soils were higher than those of healthy rhizosphere soils (*Meng et al., 2019*). As the samples were collected from an Orah orchard where fertilization was a common practice, the greater amount of nutrients in the soil of *M. panyuensis*-infected Orah compared to that of healthy plants may be due to the atrophy of the roots of diseased plants, which are unable to take up nutrients from the soil. Furthermore, organic matter, TN, AN, TP, AP, TK, and AK were positively correlated with the bacterial communities *Burkholderia-Caballeronia-Paraburkholderia*, *Bacillus*, and *Sphingomonas* and negatively correlated with *Acidothermus*, *norank_f_norank_o_Vicinamibacterales*, *norank_f_Xanthobacteraceae*,

*norank_f_norank_o_Acidobacteriales* and *norank_f_norank_o_Gaiellales* in the *M. panyuensis*-infected Orah rhizosphere soil. Organic matter, TN, AN, TP, AP, TK, and AK were all positively correlated with the fungal communities *Lycoperdon*, *Fusarium*, *Neocosmospora*, *Talaromyces*, and *Tetragoniomyces*, and negatively correlated with *Roussoella*, *Penicillium*, *Chaetomium*, and *Acrocalymma* in the *M. panyuensis*-infected Orah rhizosphere soil.

Furthermore, the composition of (and changes in) rhizosphere microorganisms and nutrients might be regulated by pathogens. A characteristic of nematode infestation is the proliferation of secondary roots caused by the constant death of infested roots.Thus, the continuous death of roots infested by *M. panyuensis* could provide greater availability of decomposing organic matter, which changed the microbial abundance in the rhizosphere. Additionally, infected plants may trigger a 'help-seeking' mechanism that actively secretes secondary metabolites to alter soil chemical properties and affect the composition of root flora. The recruited flora may also form a competitive mechanism for better adaptation to the environment, resulting in the enrichment of specific species in the rhizosphere of diseased plants (*Hou et al., 2023*; *She et al., 2017*; *Sui et al., 2019*). All these factors change the relative abundance of rhizosphere soil microbial communities.

## CONCLUSION

The root-knot nematode responsible for disease in Orah has been identified as *M. panyuensis*. Further analysis of soil chemical properties and the microbiome revealed significant differences in organic matter, TN, AN, TP, AP, TK, and microbial community composition between the *M. panyuensis* infected Orah rhizosphere soil and healthy Orah rhizosphere soil, with notable correlations observed. A range of potential biocontrol strains has been identified to enhance the diversity of microbial biocontrol agents, particularly *Burkholderia* spp., which were isolated in this study. Preliminary investigations indicate that this *Burkholderia* spp. exhibits effective control against various plant pathogens, including *F. oxysporum*, *Ralstonia solanacearum*, and *Meloidogyne enterolobi*i. These findings may inform the screening of biocontrol strains and provide a scientific foundation for the prevention and treatment of root-knot nematodes, and other diseases.

### Funding

This research supported by the National Natural Science Foundation of China (32202245), the Specific Research Project of Guangxi for Research Bases and Talents (Guike AD23026066, Guike AD23026099), and the China Agriculture Research System-Guangxi Citrus Innovative Research Team (nycytxgxcxtd-2021-05-03). The funders had no role in study design, data collection and analysis, decision to publish, or preparation of the manuscript.

## Grant Disclosures

The following grant information was disclosed by the authors:
National Natural Science Foundation of China: 32202245.
Specific Research Project of Guangxi for Research Bases and Talents: Guike AD23026066, Guike AD23026099.
China Agriculture Research System-Guangxi Citrus Innovative Research Team: nycytxgxcxtd-2021-05-03.

## Competing Interests

The authors declare that they have no competing interests.

## Author Contributions

- Xiaoxiao Zhang conceived and designed the experiments, performed the experiments, authored or reviewed drafts of the article, and approved the final draft.
- Wei Zhao performed the experiments, analyzed the data, prepared figures and/or tables, and approved the final draft.
- Yuming Lin performed the experiments, prepared figures and/or tables, and approved the final draft.
- Bin Shan performed the experiments, authored or reviewed drafts of the article, and approved the final draft.
- Shanshan Yang conceived and designed the experiments, performed the experiments, analyzed the data, prepared figures and/or tables, authored or reviewed drafts of the article, and approved the final draft.

## Data Availability

The data is available at NCBI: PRJNA1010647. Bacterial microbiomes of the healthy Orah rhizosphere soil were SRR25907662, SRR25907661 and SRR25907658; Bacterial microbiomes of the root-knot-infected Orah rhizosphere soil were SRR25907657, SRR25907656 and SRR25907655; Fungal microbiomes of the healthy Orah rhizosphere soil were SRR25907654, SRR25907653 and SRR25907652; Fungal microbiomes of the root-knot-infected Orah rhizosphere soil were SRR25907651, SRR25907660 and SRR25907659.

## Supplemental Information

Supplemental information for this article can be found online at http://dx.doi.org/10.7717/peerj.18495#supplemental-information.

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
