# Peer review of "Identification of Meloidogyne panyuensis (Nematoda: Meloidogynidae) infecting Orah (Citrus reticulata Blanco) and its impact on rhizosphere microbial dynamics: Guangxi, China"

_PeerJ, doi:10.7717/peerj.18495_

## Round 0.1 · original submission · Major Revisions

Please revise the manuscript according to the reviewers' comments.

Reviewer 1 ·

Basic reporting

The article reports the presence of M. panyuensis infected Orah in a production area in China, and data comparing chemical and biological characteristics of the rhizosphere of infested and non-infested plants. Although the data are completely descriptive, the nematode identification work and metagenomics provide valuable information to direct control efforts for this pest nematode. Some inconsistencies in the summary and introduction are pointed out in the attached file. I propose a new wording to establish the objectives at the end of the introduction.

Experimental design

The information on the collection sites is not available.
It is mandatory to describe how many sites/orchards were studied, how many samples per site and the sampling design (zig-zag, transect, etc.).
It is also mandatory to describe the state of the plantations where it was collected. Type of crop, age of the plants, agronomic management activities.
Were samples from plants infected with nematodes taken in the same orchards as samples from healthy plants?
Were the diseased plants and healthy plants subjected to the same agronomic management scheme?
If this was not the case, the samples from healthy and diseased plants are not comparable because the chemical and biological differences in the rhizosphere were not given solely by the nematode infestation.

Except for some gaps marked in the text, the remaining methodology is well explained.

Validity of the findings

The results obtained are interesting and denote the hard work carried out by the authors. However, it is necessary for the authors to consider some aspects to improve their discussion:

1) In recent literature, it has been shown that infestation by root-knot nematodes causes severe diseases due to their association with other microorganisms; these pathobiomes are known as Meloidogyne-based diseases complex. Although with the results obtained the authors cannot affirm that this is the case in Orah, information about these pathobiomes could help interpret their results.

2) The greater microbial abundance in the rhizosphere of diseased plants can be explained by the greater availability of decomposing organic matter given by the constant death of roots infested by nematodes. A characteristic of nematode infestation is the proliferation of secondary roots caused by the constant death of infested roots, which forces the plant to replace them. The authors do not address this issue in the discussion and this could explain the greater abundance but not greater diversity compared to healthy plants.

3) The greater amount of nutrients in the soil of diseased plants compared to healthy plants may be due to the atrophy of the roots of diseased plants that are not able to take up nutrients from the soil. Furthermore, if we assume that the samples were taken in a crop where fertilization is a common practice (especially if the plants show delayed growth and symptoms of nutritional deficiencies as in the case of plants infested with nematodes), this can explain the greatest amount of nutrients in the soil of diseased plants. The authors should clarify this.

Additional comments

No comment

Annotated reviews are not available for download in order to protect the identity of reviewers who chose to remain anonymous.

·

Basic reporting

This manuscript is relatively smooth, fully explaining the performance differences of Orah after being damaged by root knot nematodes and the healthy plants. The cited references can express the current research progress and have a high degree of correlation. The structure of the paper is reasonable, and the tables and graphics used are concise and clear.
There are some issues with missing page numbers or article IDs in the references, and non-standard writing of journal names. Non English journals cited in the article should be labeled with "xxx with English abstract".

Experimental design

The specifics of the experimental design, sample sizes, and analytical techniques may vary depending on the available resources and research objectives. It is essential to consult relevant scientific literature and adapt the design accordingly.
The methodology, research objectives, sample size, data collection methods, statistical analysis of this manuscript is quite well. In this section Sample collection, I think that authors should give more details.

Validity of the findings

In this study, the researchers compared the microbial composition in the rhizosphere (the soil surrounding the roots) of healthy Orah plants with those infected by M. panyuensis, a type of root-knot nematode. By doing so, they were able to identify potential strains for controlling these nematodes. This discovery is significant as it provides a basis for developing strategies to detect and prevent root-knot nematode disease in Orah plants at an early stage. By understanding the differences in the microbial composition between healthy and infected plants, researchers and farmers can take proactive measures to protect Orah crops from this destructive disease.

Reviewer 3 ·

Basic reporting

No comment

Experimental design

Require improvement.

Validity of the findings

No comment

Additional comments

The manuscript has lots of good quality work with interesting objectives and approaches to identify microbial communities associated with the root-knot nematode of Orah. However, certain questions and issues need to be addressed in the manuscript;
Abstract
L16: Add the scientific name of Orah. As per information mentioned in the texts, Orah is a variety of citrus and it is really difficult to understand for the international audience.
L20: molecular biological method?
L30-31: Potential root-knot nematode control strains were identified. Please specify.
Keywords
Change the keywords already mentioned in the title.
Introduction
The background is too much. Please reduce it into small paragraphs by rewriting and removing unnecessary things.
L45: Please add the scientific name of Orah and its origin. This information is necessary for those who are unaware of Orah.
L49: hm2? write full form when used first.
L50: better to convert in USD or $ mentioned in the bracket.
L58: Add authority name of Meloidogyne panyuensis.
L78-79: The statement is not correct. See the following reference
Manel Labiadh, Besma Mhamdi, Ameni Loulou & Kallel Sadreddine (2023) Impact of rhizobacteria community of citrus root on Tylenchulus semipenetrans and on Citrus plant growth, Biocontrol Science and Technology, 33:3, 241-257, DOI: 10.1080/09583157.2023.2175785
L86-88: This sentence should be moved to the first paragraph of the introduction about Orah.
L88-89: Repetition.
Materials and methods:
L111: Females were considered for morphological study and J2s were used for molecular characterization, Why?
L140: Where? in orchard or pot, not clear. The methodology on plant culture is very poorly described. How was control maintained? No details on the number of plants, spacing between plants, duration, replication, etc.
Results
L204: diameter of what?
Discussion
Overall, the discussion is poor and lacks explanations and comparisons. It is better to remove the subheadings rather than use separate paragraphs.
L391-393: Not clear

---

## Round 0.2 · Minor Revisions

Dear Author

One of the reviewers raised a minor single comment "The final manuscript should add the authority name of Meloidogyne panyuensis Liao, Yang, Feng & Karssen, 2005". Kindly revise it.

Some other remarks are provided below. Please make the following revisions to the manuscript:

1) Please modify the title to make it clear that:
Meloidogyne panyuensis is a nematode, and Orah is a mandarin/citrus (Citrus reticulata) species.

Also, in the Abstract, you should provide this background (nematode/citrus) for the non-expert reader.

2) The writing must be improved. Besides various grammar errors that make the text difficult to understand, phrases such as "abundant labor resources" are not usual. Please provide evidence for professional proofreading before the article may be acceptable.

3) The authors could also stress more clearly the possible biocontrol with Burkholderia spp. (e.g., in the Abstract and Conclusions.

with regards

Reviewer 1 ·

Basic reporting

Dear Editor, Dear Authors
I have reviewed the authors' response letter and the corrected document. I agree with the changes made based on my observations.

I consider the article suitable for publication.

Experimental design

I agree with the changes made in this section

Validity of the findings

I agree with the changes made in this section

Additional comments

No additional comments

·

Basic reporting

No comment

Experimental design

No comment

Validity of the findings

No comment

Additional comments

No addtional comment

Reviewer 3 ·

Basic reporting

No comment

Experimental design

No comment

Validity of the findings

No comment

Additional comments

The final manuscript should add the authority name of Meloidogyne panyuensis Liao, Yang, Feng & Karssen, 2005.

---

## Round 0.3 · Major Revisions

The original Academic Editor is not available so I have taken over handling this submission.

The study requires several significant revisions, primarily in defining its objectives and rationale more clearly. The introduction would benefit from substantial improvements in English language usage. Additionally, the discussion section deviates excessively from the main topic and does not align closely with the study's objectives and findings. Please refer to my detailed comments in the attached PDF file. Upon reviewing your responses to these comments, I will be better positioned to make an informed decision.

---

## Round 0.4 · Minor Revisions

The reviewers recommend the acceptance of the paper, and I agree that the study was well performed, and the manuscript can be published after:

1) Professional proofreading. E.g., the very first sentence of the abstract already contains the first grammar error: "Root-knot nematode disease severely affect the yield and quality of the mandarin variety (Citrus reticulata Blanco “Orah”) in Guangxi, China." => must be 'affects' (Pl.), and the species/variety name should not be in parenthesis.

2) The conclusions should be revised and extended. The impact on practical applications should be made more clear.

·

Basic reporting

no comment

Experimental design

no comment

Validity of the findings

no comment

Additional comments

no comment

Reviewer 3 ·

Basic reporting

No comments

Experimental design

No comments

Validity of the findings

No comments

Additional comments

No comments

---

## Round 0.5 · accepted · Accept

Thank you very much for proofreading your manuscript! I appreciate your effort and your contribution to the community.